# The Role of Small Extracellular Vesicles in the Progression of Colorectal Cancer and Its Clinical Applications

**DOI:** 10.3390/ijms23031379

**Published:** 2022-01-26

**Authors:** Li-Chun Chang, Han-Mo Chiu, Ming-Shiang Wu, Tang-Long Shen

**Affiliations:** 1Department of Internal Medicine, National Taiwan University Hospital, Taipei 100, Taiwan; lichunchang@ntu.edu.tw (L.-C.C.); hanmochiu@ntu.edu.tw (H.-M.C.); mingshiang@ntu.edu.tw (M.-S.W.); 2Health Management Center, National Taiwan University Hospital, Taipei 100, Taiwan; 3Department of Plant Pathology and Microbiology, National Taiwan University, Taipei 100, Taiwan; 4Center for Biotechnology, National Taiwan University, Taipei 100, Taiwan; 5Genome and Systems Biology Degree Program, National Taiwan University, Taipei 100, Taiwan

**Keywords:** colorectal cancer, small extracellular vesicle, blood test, biomarker

## Abstract

Colorectal cancer (CRC) is one of the most common cancers worldwide and a longstanding critical challenge for public health. Screening has been suggested to effectively reduce both the incidence and mortality of CRC. However, the drawback of the current screening modalities, both stool-based tests and colonoscopies, is limited screening adherence, which reduces the effectiveness of CRC screening. Blood tests are more acceptable than stool tests or colonoscopy as a first-line screening approach. Therefore, identifying blood biomarkers for detecting CRC and its precancerous neoplasms is urgently needed to fulfill the unmet clinical need. Currently, many kinds of blood contents, such as circulating tumor cells, circulating tumor nucleic acids, and extracellular vesicles, have been investigated as biomarkers for CRC detection. Among these, small extracellular vesicles (sEVs) have been demonstrated to detect CRC effectively in recent reports. sEVs enable intercellular shuttling—for instance, trafficking between recipient cancer cells and stromal cells—which can affect tumor initiation, proliferation, angiogenesis, immune regulation; metastasis, the cancer-specific molecules, such as proteins, microRNAs, long noncoding RNAs, and circular RNAs, loaded into cancer-derived sEVs may serve as biomarkers for the detection of cancers, including CRC. Indeed, accumulating evidence has shown that nucleic acids and proteins contained in CRC-derived sEVs are effective as blood biomarkers for CRC detection. However, investigations of the performance of sEVs for diagnosing CRC in clinical trials remains limited. Thus, the effectiveness of sEV biomarkers for diagnosing CRC needs further validation in clinical trials.

## 1. Introduction

Colorectal cancer (CRC) is one of the most common cancers worldwide. To face the challenge posed by the current CRC prevalence, nationwide screening programs have been implemented worldwide. Structural screening has been proven effective in reducing the incidence and mortality of CRC [1,2]. The current modalities for CRC screening can be divided into the following two major categories: stool-based and direct visualization tests. Stool-based tests include fecal immunochemical tests (FITs), guaiac fecal occult blood tests, and stool DNA tests. On the other hand, direct visualization tests include colonoscopy, flexible sigmoidoscopy, and computed tomography colonoscopy. However, both stool-based and direct visualization tests have the drawback of limited screening adherence. Approximately one-third of eligible subjects aged 50 to 75 years have never been screened for CRC, and half are inadequately screened [3]. Thus, it becomes critical to increase adherence and subsequent improvements in the effectiveness of CRC screening. Accordingly, blood tests could increase participation by 6.5% participation compared with FITs for CRC screening [4]. Not surprisingly, there is a preference for blood-based screening to enhance the suboptimal acceptance of either colonoscopy or stool-based tests due to the desire for convenience, inexpensiveness and social etiquette, highlighting the need for and potential merit of developing highly robust blood-based tests for CRC detection.

Cancer embryonic antigen (CEA) is an attractive complementary tool for CRC diagnosis. However, the poor sensitivity and specificity of CEA limit its application in the detection of CRC. Accordingly, researchers have made great efforts to identify reliable and accessible biomarkers in the blood for decades, first focusing on circulating tumor cells (CTCs) released from tumors and then on circulating tumor-related nucleic acids (mainly DNA and miRNA). Given that they are vastly outnumbered by other cells, especially white blood cells, CTCs are present in blood in minimal quantities. Thus, their detection requires an isolation/enrichment step before collection. Because of their sparse distribution in blood, the main drawback of CTCs is that their analysis requires extremely sensitive and specific methods and often needs isolation/enrichment [5]. Circulating tumor DNA (ctDNA) is DNA released from necrotic tumor cells that then circulates in the peripheral blood [6]. The concentration of ctDNA depends on the type and stage of the tumor, usually increasing in more advanced stages of cancer [7]. Therefore, blood tests using ctDNA are incapable of detecting precancerous lesions and early stage cancers [8,9]. Small extracellular vesicles (sEVs) were initially recognized as cellular debris and considered to dispose of cellular components when first discovered and coined “exosomes” in 1983 [10,11]. The molecules contained in sEVs can be shuttled between recipient cancer cells and stromal cells, thereby regulating the tumor microenvironment by modulating various signaling pathways and functions. In the last decade, increasing interest has focused on the participation of circulating sEVs in tumor proliferation, metastasis, angiogenesis, tumor immunity, and drug resistance in the field of cancer biology. On the other hand, cancer-specific molecules, such as proteins, microRNAs (miRNAs), long noncoding RNAs (lncRNAs), and circular RNAs (circRNAs), loaded into cancer-derived sEVs may serve as biomarkers for the detection of cancers, including CRC. This review highlights the oncogenic roles of cancer-derived sEVs in CRC carcinogenesis and their clinical applications in CRC diagnosis.

## 2. Small Extracellular Vesicles in Cancer Communication

sEVs are defined as a specific subset of extracellular vesicles that range from 30 to 150 nm in size. They originate as endosomal-derived intraluminal vesicles (ILVs) that are subsequently released into the extracellular milieu by the fusion of multivesicular endosomes or multivesicular bodies (MVBs) with the intracellular side of the plasma membrane. sEV-mediated intercellular communication plays a critical role in cancer development. The selective packaging of oncogenic molecules renders tumor-derived sEVs capable of altering the tumor microenvironment, modulating tumor immunity, and promoting cancer development, undoubtedly contributing to cancer recurrence and drug resistance. Notably, the molecular and functional characteristics of sEVs evolve with cancer progression. Thus, tumor-derived sEVs can provide valuable real-time information to reflect the dynamic changes in cancer progression. Given the unique molecular profiles and signatures, the cargo of tumor-derived sEVs has been highlighted as a useful diagnostic and predictive biomarker [12].

sEVs regulate the molecular and functional communication between cancer cells and their tumor microenvironment during cancer progression [13]. A previous study found that breast cancer cells secreting sEVs containing CXCR4 (CXC motif chemokine receptor 4) exhibited high levels of stemness-related markers and metastasis-related mRNAs, which enhanced their invasive ability and metastatic potential [14]. In addition, adipose-derived mesenchymal stem cells secrete sEVs to facilitate cancer migration and proliferation via a pathway independent of the Wnt-β-catenin pathway [15]. In CRC, sEVs derived from cancer-associated fibroblasts (CAFs) have been shown to prime cancer stem cells and contribute to chemoresistance through Wnt signaling [16]. These studies indicated the essential role of sEVs in intercellular communication during cancer progression. Tumor-derived sEVs with tumorigenic activity regulate cancer progression through multiple perspectives, including cancer aggressiveness, cancer invasiveness, extracellular matrix remodeling, angiogenesis, drug resistance, and immunosuppression [17,18], suggesting the essential implications of tumor-derived sEVs on cancer development and cancer therapy.

## 3. Small Extracellular Vesicles in the Proliferation, Invasion, and Migration of CRC Cells

The canonical Wnt-β-catenin pathway is constitutively active during CRC development, and mutations that involve the Wnt signaling cascade are the most prevalent genetic alterations in CRC carcinogenesis [19]. sEVs recruit mutant β-catenin and activate the Wnt signaling pathway, resulting in stimulation of the proliferation and migration of recipient cells [20]. Overexpression of CAF-derived sEV-miR-92a-3p downregulates the expression of its target genes FBXW7 (F-box and WD repeat domain containing 7) and MOAP1 (modulator of apoptosis 1) to inhibit the ubiquitination and degradation of β-catenin, resulting in the invasion and migration of CRC cells via activation of the Wnt-β-catenin signaling pathway [21]. At present, an increasing number of studies are finding that sEV-derived molecules, such as lncRNAs and circRNAs, are involved in specific signaling axes to alter tumor metastasis [22,23,24].

## 4. Small Extracellular Vesicles and Epithelial-Mesenchymal Transition in CRC

Epithelial-mesenchymal transition (EMT) is considered one of the most crucial mechanisms required for tumor metastasis [25] and is characterized by a loss of epithelial properties and gain of mesenchymal properties. Therefore, the presence of EMT is associated with a poor prognosis [26]. Growing evidence has revealed the relationship between sEVs and EMT. Recent studies have found that tumor-derived sEVs can contribute to the EMT process in recipient cells [27]. sEVs secreted by adherent cells were taken up by metastatic cells, subsequently initiating EMT and metastasis in the recipient cancer cells. A previous study found that the level of miR-210 in sEVs was significantly higher than intracellular level in metastatic cancer cells. The upregulated miRNA-210 in sEVs was suggested to trigger the EMT process, contributing to a metastatic phenotype through a coordinated tumor–tumor microenvironment cascade [28]. Many miRNAs, such as miR-106b-3p, miR-25-3p, miR-130b-3p, and miR-425-5p, may also promote EMT through different pathways [29,30]. Additionally, circRNAs in the sEV might function as competing endogenous RNAs in the regulation of EMT in patients with metastatic CRC.

## 5. Small Extracellular Vesicles and the Premetastatic Niche of CRC

Compelling evidence indicates that tumor cells influence other organs and tissues before metastasizing. These newly emerging locations are known as premetastatic niches (PMNs). Based on Steven Paget’s “seed-and-soil” theory, the novel concept of the PMN was proposed to elucidate the mechanisms of tumor metastasis from the primary site to the metastatic site [31]. The PMN is a microenvironment in which tumor-derived cytokines, growth factors, immune cells, and sEVs are involved to prepare an optimal site for the seeding and growth of metastatic cancer cells [32].

PMNs possess the following six characteristics to promote metastasis and enable tumor growth: vascular leakiness and angiogenesis, lymphangiogenesis, inflammation, immunosuppression, genetic reprogramming, and organotropism [33]. sEVs play a vital role in PMN formation and participate in each of the above characteristics (Figure 1). CRC-derived sEVs containing miR-25-3p contribute to the induction of vascular leakiness and angiogenesis by downregulating the expression of the Krüppel-like factor family. Furthermore, sEVs containing miR-25-3p may impair the junctions in the endothelial cell layer, which increases the formation of PMNs and CRC metastases in the liver and lung [34,35]. Lymphangiogenesis is actively involved in the formation of the PMN. CRC-derived sEVs containing IRF-2 (interferon regulatory factor 2) have been postulated to stimulate VEGF-C (vascular endothelial growth factor C) secretion, resulting in lymphangiogenesis and metastasis [36]. Chronic inflammation is a critical driver of tumor progression and metastasis, and thus, the local inflammatory microenvironment is an essential factor contributing to the formation of the PMN. CRC-derived sEVs containing miR-21 help polarize liver macrophages into an IL-6-secreting phenotype by binding to TLR7 (Toll-like receptor 7) in tumor-associated macrophages, contributing to the inflammatory environment [37]. A hallmark of PMN formation is the recruitment of immune cells to establish an immunosuppressive microenvironment. Consistent with this observation, bone marrow-derived dendritic cells (BMDCs) are essential effector cells that suppress the antitumor response and allow primary tumor cells to escape immune defense [38]. Likewise, CRC-derived sEVs containing miR-1246 may result in macrophage polarization into M2 tumor-associated macrophages and subsequently contribute to the recruitment of BMDCs and establishment of an immunosuppressive microenvironment [39]. Stromal, metabolic, and epigenetic reprogramming contributes to PMN-induced tumor metastasis. Previous studies have demonstrated that sEVs are associated with the recruitment and reprogramming of host stromal cells, such as fibroblasts and epithelial cells, to modify the PMN [40]. Organotropism is defined as the preference of certain tumors to metastasize to specific organs. Tumor-derived sEVs express specific integrins (ITGs), which interact with extracellular matrix molecules to initiate the formation of PMNs in a particular tissue [41]. Integrin α6/integrin β4/integrin β1 and integrin β5/integrin αv are enriched in sEVs with lung and liver tropism, respectively. Expression of the specific integrin contained in sEVs may likewise orchestrate organotropism in CRC [31]. Taken together, recent studies have provided solid evidence to support the vital role of sEVs in the formation of PMNs in CRC, and clarification of the mechanism may offer a window for the early diagnosis and prevention of CRC metastasis.

## 6. Small Extracellular Vesicles and Immune Regulation in CRC

Cancer-derived sEVs may impair immune cell maturation and antitumor activity, thereby establishing an immunosuppressive microenvironment and promoting the immune escape of cancer cells [42]. Cancer cells may present programmed cell death ligand-1 (PD-L1) on their surface and then attenuate antitumor immunity by binding the programmed cell death protein-1 receptor on effector T cells. PD-L1^+^ sEVs were also capable of inhibiting T cell activity in CRC cells, promoting tumor growth, and resisting immune checkpoint protein inhibitors [43]. CD8^+^ T cells act as antitumor effector cells in the tumor microenvironment, and the dysfunction of CD8^+^ T cells impairs the immune system against cancer [44]. sEVs containing miR-146a-5p were shown to decrease the number of tumor-infiltrating CD8^+^ T cells, thereby establishing an immunosuppressive microenvironment in CRC [45].

## 7. Small Extracellular Vesicles as Biomarkers for Cancer Diagnosis

Given the vast number of molecular cargoes, several studies have shown that each category of sEV cargo in different cancer cell lines and patient blood samples includes many potential biomarkers for cancer diagnosis, compared with the cargo in their corresponding normal counterparts. In the past, many potential noninvasive biomarkers via blood or urine tests have been developed for cancer detection; many of these biomarkers resided within sEV [46,47]. The enrichment and stability of these sEV biomarkers may achieve better diagnostic performance [48,49,50]. The current position of each sEV cargo as a biomarker for cancer diagnosis is addressed as follows:

### 7.1. Protein Biomarkers in sEVs

Among versatile molecules in sEVs, proteins are the first component to be investigated. The biological role of the sEV proteome has been disclosed in CRC [35,51,52], bladder cancer [53], prostate cancer [54], pancreatic cancer [55], lung cancer [56], breast cancer [57], hepatoma [58], hematopoietic cancer [59], ovarian cancer [60], and buccal squamous cell carcinoma [61]. sEV proteins from cancer cells affect the tumor microenvironment through suppressive modulation of immune surveillance and immune cells, including NK cells, T cells, and macrophages [62]. Large-scale proteomic analysis of sEVs allowed the discovery of novel markers and signatures across varied tumor types. Moreover, a previous study demonstrated unique proteomic profiles in sEVs across distinct cancer types, including CRC [63]. Further establishment of a reliable and reproducible isolation procedure and analysis protocol will be essential to classify the sEV proteome as a novel modality for cancer diagnosis.

### 7.2. Noncoding microRNAs in sEVs

miRNAs are the most extensively studied category of short noncoding RNAs because of their profound regulatory function in gene expression. The role of different sEV-derived miRNAs in cancer has been explored in numerous studies [64]. Briefly, sEV-derived miRNAs can control the proliferation, invasion, and metastasis of recipient tumor cells. sEV-derived miRNAs mediate widespread communication between tumor cells and various cells in their environment, such as CAFs, immune cells, and endothelial cells. Moreover, drug-resistant tumor cells can transmit the resistance phenotype to drug-sensitive tumor cells through horizontal transfer of sEVs containing miRNAs [65]. Regarding cancer diagnosis, previous systemic reviews have explored the role of sEV-derived miRNAs in detecting non-small-cell lung cancer [66], pancreatic cancer [67], hepatoma [68], lymphoma [69], glioma [70], CRC [71], breast cancer [72] and nasopharyngeal cancer [73] as biomarkers.

### 7.3. Long Noncoding RNAs in sEVs

sEVs containing long RNAs, ranging from mRNAs, circRNAs to lncRNAs, can also be reliably found and detected in plasma and reflect their tissue origins. A previous study explored whether sEVs containing long RNAs may serve as biomarkers for hepatoma detection [74]. lncRNAs are a subclass of noncoding RNAs more than 200 nucleotides in length that are closely related to the development of many types of cancer. lncRNAs can be packed into sEVs, and tumor cells can secrete specific lncRNAs in sEVs via a yet-unknown mechanism. Dysregulated lncRNAs have been reported to be involved in regulating the proliferation, metastasis, and recurrence of multiple cancers, including gastric cancer [75], CRC [76], lung cancer [77], prostate cancer [78], hepatoma [79], cervical cancer [80], glioma [81], bladder cancer [82], renal cancer [83], cholangiocarcinoma [84] and ovarian cancer [85]. Furthermore, sEV-derived lncRNAs mediate the progression and chemoresistance of tumor cells in the tumor microenvironment in many cancers. Thus, sEV-derived lncRNAs may play a role in cancer diagnosis as biomarkers.

### 7.4. Circular RNAs in sEVs

CircRNAs exist in eukaryotes and are generated through backsplicing, a specific form of alternative splicing, and they are more stable than linear RNAs. CircRNAs influence cancer cell proliferation, invasion, metastasis, and chemoresistance, and they work upon sEVs reaching either neighboring or distal cells. A previous study demonstrated that deregulated sEV-derived circRNAs were more highly expressed in subjects with metastatic and localized breast cancer than in healthy controls [86]. Moreover, dysregulated expression of sEV-derived circRNAs was also noted in gastric cancer [87], CRC [88], pancreatic cancer [89], hepatoma [90], cholangiocarcinoma [91], small-cell lung cancer [92], and urogenital cancer [93]. However, the low abundance of circRNAs in sEVs makes their detection challenging. Thus, better quantitative approaches should be established to accurately detect the expression level of sEV-derived circRNAs.

As mentioned above, the diverse cargos in sEVs have demonstrated potential for detecting different cancers. Among the multiple cargos carried by sEVs, it is critical to clarify which molecule is the optimal diagnostic biomarker in terms of sensitivity, specificity, and accuracy. Moreover, it needs to be defined which cancer may benefit from the application of a certain sEV-derived molecule for diagnosis. Wong et al. conducted a meta-analysis to review 30 studies investigating sEV-derived molecules as diagnostic biomarkers [94]. More than half of these studies focused on digestive organ cancers, including four studies in CRC, five studies in liver cancer, four in pancreatobiliary cancer, and four in gastric cancer. A total of 47 diagnostic biomarkers were enrolled in the meta-analysis, and 42.6% of the biomarkers were miRNAs, followed by lncRNAs (36.2%) and proteins (19.1%). Among these biomarkers, sEV-miR-21 was the main biomarker investigated for diagnosing various cancers. Regarding the kind of specimen for collecting sEVs, 61.3% of the sEV molecules were collected from serum, followed by plasma (16.1%), urine (12.9%), saliva (3.2%), and bile (3.2%).

This meta-analysis also explored the sensitivity and specificity of sEV biomarkers for detecting various cancers [94]. The authors focused on CRC (four studies with eleven biomarkers), gastric cancer (four studies with five biomarkers), pancreatic cancer (four studies with eight biomarkers), liver cancer (four studies with seven biomarkers), and prostate cancer (four studies with seven biomarkers). The diagnostic sensitivity and specificity of the pooled sEV-derived biomarkers were 0.57 and 0.87 for detecting CRC, 0.77 and 0.73 for gastric cancer, 0.91 and 0.90 for pancreatic cancer, 0.76 and 0.80 for liver cancer, and 0.77 and 0.79 for prostate cancer, respectively. The pooled sensitivity and specificity of sEVs were optimal and could effectively distinguish cancer patients from their noncancerous counterparts.

## 8. Application of Small Extracellular Vesicles as Biomarkers for CRC Detection

sEVs contain molecules that can be incorporated into the development of novel tests for detecting precancerous colorectal neoplasms and cancer, which enable them to be applied in diagnostic tests. Currently, miRNAs are the most commonly investigated molecules in sEVs for CRC detection. In previous studies, the utility of sEV-derived miRNAs has been tested for CRC diagnosis. The serum levels of sEV-derived miRNAs (miR-1229, miR-1224-5p, miR-223, let-7a, miR-150, and miR-21) were significantly higher in CRC patients than in controls, and their positive rates were 22.7%, 31.8%, 46.6%, 50%, 55.7%, and 61.4%, respectively. The false-positive rates of these miRNAs ranged from 0% to 9% [95]. Compared with CEA and CA19-9, which had sensitivities of 30.7% and 16%, respectively, sEV-derived miRNAs exhibited superior performance and might be useful as new diagnostic biomarkers. Moreover, the expression of miR-19a-3, miR-21-5p, and miR-425-5p in sEVs was also higher in serum from subjects with CRC than in that from normal controls; thus, these miRNAs could facilitate CRC detection as biomarkers [96]. In addition to facilitating diagnosis, sEV-derived miRNAs may play a role as prognostic biomarkers in predicting CRC recurrence. The expression of miR-17-92 in sEVs was significantly higher in CRC patients with poor prognosis than in healthy counterparts and thus had been proposed as a biomarker of CRC recurrence [97]. sEVs containing miR-122 are another potential biomarker for predicting the survival outcome of CRC because the elevated expression of miR-122 in sEV is closely associated with the presence of liver metastasis [98].

In addition to miRNAs, lncRNAs are another sEV molecule commonly investigated for CRC diagnosis. The serum level of sEVs containing lncRNA-CCAT2 (colon cancer associated transcript 2) was increased in cancer tissue and serum from CRC patients. The serum level of sEVs containing lncRNA-CCAT2 was higher in advanced CRC and decreased markedly after surgery, thus identifying this lncRNA as a good candidate for monitoring CRC as a biomarker [99]. Moreover, sEVs containing lncRNA-XIST (X-inactive specific transcript) presented convincing performance for CRC detection [100]. CircRNAs are closely associated with the initiation and development of cancers and play a role in CRC detection. Thus, sEVs containing hsa-circ-0004771 originate from the tumor, and their level is significantly increased in the serum of CRC patients, thereby identifying this circRNA as a potential biomarker for the early diagnosis of CRC [101]. The proteins expressed on the surface of sEVs are also promising candidates for CRC detection. Full-length cadherin-17 [102], CD147 [103], cellular prion protein [104], GCLM (glutamate-cysteine ligase modifier subunit), KEL (Kell metallo-endopeptidase), APOF (apolipoprotein F), CFB (complement factor B), PDE5A (phosphodiesterase 5A), and ATIC (5-aminoimidazole-4-carboxamide ribonucleotide formyltransferase/IMP cyclohydrolase) [105] were found to be secreted by specific CRC-derived sEVs and therefore may potentially be biomarkers for diagnosing CRC in the future. Furthermore, it is worth clarifying whether these promising biomarkers, such as miRNAs in combination with surface proteins, may further improve the diagnostic performance and effectiveness of CRC detection.

sEVs, as vehicles of intercellular communication, are increasingly close to being recognized as biomarkers for the diagnosis of CRC. Nucleic acids and proteins contained in CRC-derived sEVs from the blood could potentially be used as biomarkers to diagnose and monitor CRC. Although a growing body of studies have explored the potential of sEV molecules as diagnostic tools, clinical trials that evaluate sEV molecules as biomarkers for CRC detection remain rare. By searching the Clinicaltrial.gov website, accessed on 28 December 2021, several clinical trials have addressed the detection of CRC by sEVs. Clinical trials conducted in China and France have investigated changes in sEV molecules before and after concurrent chemoradiation therapies for rectal cancer (NCT03874559 and NCT04227886). Two clinical trials evaluated the performance of sEV proteins and miRNAs for the early diagnosis of CRC (NCT04394572 and NCT04523389, respectively). The results of these trials will help clarify the performance of sEVs for the early diagnosis and prediction of treatment response in CRC. To date, no clinical trial has investigated the effectiveness of blood tests with sEV molecules for screening purposes, and further study of this issue is warranted.

## 9. Conclusions

The survival outcome of CRC largely depends on the cancer stage at diagnosis, and early diagnosis markedly improves prognosis and survival. Effective screening reduces CRC mortality by detecting cancer at an asymptomatic and early stage. However, the attendance rate is one of the most critical factors in accomplishing the CRC screening program. Both stool-based and primary colonoscopy screening program have the limitation of not being broadly accepted by the participants. Undoubtedly, a blood test is the most acceptable approach for screening and has the potential to increase screening attendance, thus reducing CRC mortality. Blood tests diagnose CRC by detecting CTCs, ctDNA, circulating RNA, sEVs, and so on. Among these molecules, sEVs are tightly associated with each stage of CRC progression, including initiation, proliferation, and metastasis, thereby providing a window to detect CRC at not only the advanced stage but also the precancerous stage. Because of their stability and feasibility for enrichment, molecules contained in sEVs are potential candidates for the development of diagnostic biomarkers. A growing number of studies have explored the diagnostic performance of sEV molecules for CRC detection. However, some challenges and difficulties related to sEVs should not be ignored. A growing number of sEV biomarkers have been identified to be specific to CRC and may be used as biomarkers to detect CRC. However, clinical trials that investigate the diagnostic performance of sEVs for CRC remain very rare. Moreover, the actual performance of sEVs for detecting colorectal neoplasias, including precancerous neoplasms and invasive cancers, in a screening setting remains unknown. Investigation of the performance of sEVs in the blood as a screening test is warranted in the future. The method of isolation and characterization of sEV varies in different studies, and standardization of these processes is critical to make the experimental results reproducible. Some non-sEV components, such as albumin, immune complexes, and platelets, may be coisolated with sEVs, which may interfere with interpreting the information from sEVs. Optimization of the isolation method is necessary to overcome this limitation. sEVs exist in many kinds of body fluids, such as blood, urine, bile, saliva, stool, and ascites. Each body fluid is supposed to play a different role in detecting various cancers. For example, sEVs in stool, bile, and saliva may potentially be used to detect CRC, bile duct cancer, and oropharyngeal/esophageal cancer, respectively. This is an exciting issue and worthy of clarification in the future. Finally, quantification of sEVs is essential to establish a cutoff for diagnosis. However, the methods for sEV quantification remain imperfect and need further improvement. The substantial implication of sEVs in each stage of CRC development has already been proven. However, the applications of sEVs as biomarkers for early diagnosis, screening, monitoring, and even predicting CRC remain immature and need to be investigated in clinical trials.

## Figures and Tables

**Figure 1 ijms-23-01379-f001:**
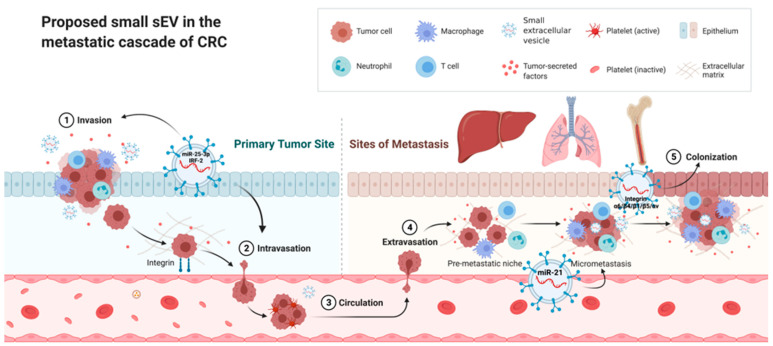
Proposed role of small EVs in the metastatic cascade of CRC Premetastatic niches (PMNs) possess six characteristics: vascular leakiness and angiogenesis, lymphangiogenesis, inflammation, immunosuppression, genetic reprogramming, and organotropism. sEVs play a vital role in each part of the above characteristics. CRC-derived sEVs containing miR-25-3p contribute to the induction of vascular leakiness and angiogenesis. Furthermore, sEVs containing miR-25-3p may impair the junctions of the endothelial cell layer. CRC-derived sEVs containing IRF-2 have been postulated to stimulate VEGF-C (vascular endothelial growth factor C) secretion, resulting in lymphangiogenesis and metastasis. sEVs containing miR-21 help polarize liver macrophages into an IL-6-secreting phenotype, contributing to the inflammatory environment. Tumor-derived sEVs express specific integrins to initiate the formation of PMNs in a particular tissue. Integrin α6/integrin β4/integrin β1 and integrin β5/integrin αv are enriched in sEVs with lung and liver tropism, respectively. (The figure was created at Biorender.com on 9 January 2022).

## Data Availability

Not applicable.

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
