# Peer review of "The Role of Small Extracellular Vesicles in the Progression of Colorectal Cancer and Its Clinical Applications"

_ijms, 2022, doi:10.3390/ijms23031379_

Round 1

Reviewer 1 Report

The paper: “The Power of Small Extracellular Vesicles  as Blood Biomarkers for Colorectal Cancer Detection” presents broad knowledge on sEV in etiopathogenesis and progression of colorectal cancer.

In my opinion the title of the paper is a little misleading- there are just few reports on the role of sEV as biomarker of CRC [71,76,88, 95-98]. These data can not be even gathered for review/ systemic review to estimate overall sensitivity and specificity. I would suggest changing the title as most of the paper quite nice describes the role of sEV in the progression of CRC and its metastatic cascade.

I would suggest major English check of the text- even in the title there is a mistake.

Some sentences are not acceptable in the language form: for instance:… However, the investigation of the sEV performance for diagnosing CRC through clinical trial remains few.

Why do some references  appear in blue font?

CA 199 should be rather called CA 19-9

I recommend major revisions.

Author Response

Q1. In my opinion the title of the paper is a little misleading- there are just few reports on the role of sEV as biomarker of CRC [71,76,88, 95-98]. These data can not be even gathered for review/ systemic review to estimate overall sensitivity and specificity. I would suggest changing the title as most of the paper quite nice describes the role of sEV in the progression of CRC and its metastatic cascade.

A1. Thank you for the comment. We have changed the title as following: “The Role of Small Extracellular Vesicles in the Progression of Colorectal Cancer and Its Clinical applications.”

Q2. I would suggest major English check of the text- even in the title there is a mistake. Some sentences are not acceptable in the language form: for instance:… However, the investigation of the sEV performance for diagnosing CRC through clinical trial remains few.

A2. Amended. We have sent the manuscript to American Journal Experts (AJE) for English editing as your valuable suggestion. The editing certificate has been provided as supplementary file and the verification code is 6313-AF10-7651-31B7-5608.

Q3. Why do some references appear in blue font?

A3. Thank you for pointing out the error. The mistake has been revised accordingly.

Q4. CA 199 should be rather called CA 19-9

A4. Thank you for the comment. We have replaced CA199 with CA19-9. (page 10, line 34)

Reviewer 2 Report

In this review authors write about the usefulness of Small Extracellular Vesicles as Blood Biomarkers for Colorectal Cancer Detection. Authors presented a comprehensive overview of the field and touched on all crucial aspects. Readers will find the review a great starting point for learn about this field. I recommend the publication of manuscripts in current form.

Author Response

Thank you so much for the kind review and encouraging comments.

Reviewer 3 Report

In this review, the authors discussed the potential use of blood small extracellular vesicles as biomarkers of colorectal cancer (CRC) diagnosis. Overall, the content of the manuscript is meaningful and important, especially for the high mortality of CRC if found at a late stage.

However, some minor revisions are required to improve the manuscript.

  1. In the Abstract, circulating tumor cells do not belong to molecules.
  2. The grammar of the sentence ‘Accordingly, researchers have been broadly implemented on developing reliable and accessible biomarkers in the blood for decades, first focused on searching for circulating tumor cells (CTCs) released from tumor lesions into the blood circulation and then circulating tumor-related nucleic acids (mainly DNA and miRNAs).’ is not correct.
  1. Abbreviations, such as FBXW7, MOAP1, and IRF-2, should be explained at the first time shown in the manuscript.
  2. Resolution of Figure 1 should be improved.
  3. References format should be changed according to the Journal.
  4. More protein markers for CRC should be discussed in the manuscript, except full-length cadherin-17.
  5. Remove references from acknowledgment.

Author Response

In this review, the authors discussed the potential use of blood small extracellular vesicles as biomarkers of colorectal cancer (CRC) diagnosis. Overall, the content of the manuscript is meaningful and important, especially for the high mortality of CRC if found at a late stage.

However, some minor revisions are required to improve the manuscript.

Q1. In the Abstract, circulating tumor cells do not belong to molecules.

A1. Thank you for the valuable correction. We have rephrased the sentence in the abstract as below: “Currently, many kinds of blood contents, such as circulating tumor cells, circulating tumor nucleic acids, and extracellular vesicles, have been investigated as biomarkers for CRC detection.” (page 3, line 9)

Q2. The grammar of the sentence ‘Accordingly, researchers have been broadly implemented on developing reliable and accessible biomarkers in the blood for decades, first focused on searching for circulating tumor cells (CTCs) released from tumor lesions into the blood circulation and then circulating tumor-related nucleic acids (mainly DNA and miRNAs).’ is not correct.

A2. Thank you for the comment. The sentence has been rephrased as below: “Accordingly, researchers have made great efforts to identify reliable and accessible biomarkers in the blood for decades, first focusing on circulating tumor cells (CTCs) released from tumors and then on circulating tumor-related nucleic acids (mainly DNA and miRNA).” (page 4, line 21)

Q3. Abbreviations, such as FBXW7, MOAP1, and IRF-2, should be explained at the first time shown in the manuscript.

A3. Amended. All the abbreviations have been explained at the first time shown as suggestion, including CXCR4 (CXC motif chemokine receptor 4) (page 5, line 22); FBXW7 (F-box and WD repeat domain containing 7) (page 6, line 7); MOAP1 (modulator of apoptosis 1) (page 6, line 7); IRF-2 (interferon regulator factor 2) (page 7, line 10); VEGF-C (vascular endothelial growth factor C) (page 7, line 11; figure legend); TLR7 (Toll-like receptor 7) (page 7, line 16).

Q4. Resolution of Figure 1 should be improved.

A4. Amended. The resolution of Figure 1 has been transformed into 300 dpi.

Q5. References format should be changed according to the Journal.

A5. Thank you for the comment. We have exchanged the reference format by the author instructions into the following format “Author 1, A.B.; Author 2, C.D. Title of the article. Abbreviated Journal Name YearVolume, page range.” Moreover, we also edited the reference by using EndNote software with the “MDPI” reference format. The reference 1 was listed below as an example:

  1. Doubeni, C.A.; Corley, D.A.; Quinn, V.P.; Jensen, C.D.; Zauber, A.G.; Goodman, M.; Johnson, J.R.; Mehta, S.J.; Becerra, T.A.; Zhao, W.K.; et al. Effectiveness of screening colonoscopy in reducing the risk of death from right and left colon cancer: a large community-based study. Gut 2018, 67, 291-298, doi:10.1136/gutjnl-2016-312712. (page 15-22)

Q6. More protein markers for CRC should be discussed in the manuscript, except full-length cadherin-17.

A6. Thank you for the suggestion. We have added more sEV protein markers into the manuscript. The relevant discussion has been rephrased as below: “The proteins presented on the surface of sEVs are also a promising candidate for CRC detection. Full-length cadherin-17[102], CD147[103], cellular prion protein[104], GCLM (glutamate-cysteine ligase modifier subunit), KEL (Kell metallo-endopeptidase), APOF (apolipoprotein F), CFB (complement factor B), PDE5A (phosphodiesterase 5A), and ATIC (5-aminoimidazole-4-carboxamide ribonucleotide formyltransferase/IMP cyclohydrolase) [105] was explored to be secreted by specific CRC-derived sEVs and, therefore, may potentially be a biomarker for diagnosing CRC in the future.” (page11, line 20)

Q7. Remove references from acknowledgment.

A7. Amended. The references have been removed from the acknowledgement and it has been rephrased as below: “We thank the staff of the Eighth Core Lab, Department of Medical Research, National Taiwan University Hospital for support during the study. This work was supported by grants from the Ministry of Science and Technology, Taiwan.” (page 14)

Round 2

Reviewer 1 Report

Thank you for revised ver.

I accept in present form.